

# A functional magnetic resonance imaging investigation of the autonomous sensory meridian response

Stephen D. Smith[1], Beverley K. Fredborg[2] and Jennifer Kornelsen[3]

[1] Department of Psychology, University of Winnipeg, Winnipeg, MB, Canada
[2] Department of Psychology, Ryerson University, Toronto, ON, Canada
[3] Department of Radiology, University of Manitoba, Winnipeg, MB, Canada

## ABSTRACT

**Background:** Autonomous Sensory Meridian Response (ASMR) is a sensory-emotional experience in which specific stimuli (ASMR "triggers") elicit tingling sensations on the scalp, neck, and shoulders; these sensations are accompanied by a positive affective state. In the current research, functional magnetic resonance imaging (fMRI) was used in order to delineate the neural substrates of these responses.

**Methods:** A total of 17 individuals with ASMR and 17 age- and sex-matched control participants underwent fMRI scanning while watching six 4-minute videos. Three of the videos were designed to elicit ASMR tingling and three videos were not.

**Results:** The results demonstrated that ASMR videos have a distinct effect on the neural activity of individuals with ASMR. The contrast of ASMR participants' responses to ASMR videos showed greater activity in the cingulate gyrus as well as in cortical regions related to audition, movement, and vision. This activity was not observed in control participants. The contrast of ASMR and control participants' responses to ASMR-eliciting videos detected greater activity in right cingulate gyrus, right paracentral lobule, and bilateral thalamus in ASMR participants; control participants showed greater activity in the lingula and culmen of the cerebellum.

**Conclusions:** Together, these results highlight the fact that ASMR videos elicit activity in brain areas related to sensation, emotion, and attention in individuals with ASMR, but not in matched control participants.

Corresponding author
Stephen D. Smith,
s.smith@uwinnipeg.ca

## INTRODUCTION

Individuals with Autonomous Sensory Meridian Response (ASMR) report experiencing tingling sensations on their scalp, neck, and shoulders—sometimes radiating down the back and limbs—during the perception of specific auditory or audiovisual stimuli (*Barratt & Davis, 2015*). These "tingles" are often accompanied by a calming feeling that many individuals report as being emotionally positive (*Poerio et al., 2018*). A previous survey research of individuals with ASMR has noted that the most common "ASMR triggers" are whispering, low-pitched repetitive noises such as tapping sounds, and videos depicting

socially intimate situations (e.g., having one's hair brushed; *Barratt & Davis, 2015*; *Barratt, Spence & Davis, 2017*). Although several studies have attempted to identify demographic or personality factors that might make an individual more likely to experience these atypical sensory experiences (*Fredborg, Clark & Smith, 2017*, *2018*; *Janik McErlean & Banissy, 2017*), much less is known about the neural underpinnings of ASMR. In the current research, functional magnetic resonance imaging (fMRI) was used to examine how the neural responses to ASMR-eliciting videos differ between individuals with ASMR and a matched control group.

Previous research has used neuroimaging techniques to identify potential differences between the brains of individuals with ASMR and matched controls. Using resting-state fMRI, *Smith, Fredborg & Kornelsen (2017)* found that the functional connectivity of the default mode network (DMN; *Raichle, 2015*; *Raichle et al., 2001*) differed between ASMR and control populations. ASMR participants showed reduced functional connectivity in the right precuneus and posterior cingulate, the left medial frontal gyrus and thalamus, and both the left and right superior temporal gyri. Increased functional connectivity was observed in right occipital and left frontal cortical areas. The fact that traditional nodes of the DMN show weaker functional connectivity in ASMR—and that brain regions outside of typical DMN nodes were recruited into the DMN—suggests that ASMR is associated with the blending of several resting-state networks in the brain. Such a finding is consistent with other studies of atypical sensory experiences including synesthesia (*Dovern et al., 2012*; *Tomson et al., 2013*), the experience of auditory hallucinations, (*Alderson-Day, McCarthy-Jones & Fernyhough, 2015*), and the subjective perceptual effects associated with hallucinogenic drugs (*Roseman et al., 2014*).

A recent psychophysiological study has taken the investigation of ASMR one step further by measuring autonomic nervous system responses *during* the ASMR experience itself (*Poerio et al., 2018*). Poerio et al. found that viewing ASMR-eliciting videos led to an increase in skin conductance responses (SCR) and a decrease in heart rate in individuals with ASMR, but not in control participants. These results are curious in that increased SCRs are typically associated with physiological arousal (*Boucsein, 1992*) whereas a slowed heart rate is associated with the opposite (*Johnston & Anastasiades, 1990*; *Shapiro et al., 2001*). The researchers concluded that these seemingly contradictory results are related to the complexity of the ASMR experience. Self-report studies has consistently shown that ASMR is associated with a feeling of calm (*Barratt & Davis, 2015*; *Fredborg, Clark & Smith, 2017*). However, the psychophysiological data suggest that ASMR is a physiologically arousing experience as well (*Poerio et al., 2018*). Importantly, this pattern of autonomic activity is distinct from the increased heart rate associated with aesthetic chills such as the *frisson* experienced during the perception of emotional musical pieces (*Benedek & Kaernbach, 2011*; *Grewe, Kopiez & Altenmüüller, 2009*). Indeed, survey research indicates that over 90% of individuals with ASMR view this experience as being different from frisson (*Fredborg, Clark & Smith, 2018*). Thus, it appears that the subjective positive emotions associated with ASMR co-occur with physiological responses in a manner that is both subjectively and biologically distinct from similar phenomena.

The purpose of the current study is to identify brain areas associated with the ASMR experience. Doing so will provide insights into the neurophysiological underpinnings of this phenomenon, thus allowing researchers to see *how* ASMR affects somatosensation and emotion rather than listing or describing those experiences. To date, only one fMRI study has measured brain activity during ASMR (*Lochte et al., 2018*). Lochte et al. presented five different 7-min ASMR-eliciting videos to 10 individuals with ASMR while they underwent an fMRI scan. A region-of-interest (ROI) analysis indicated that the experience of ASMR tingles was associated with increased activity in the nucleus accumbens, dorsal anterior cingulate gyrus, supplementary motor area, and a region including the insula and inferior frontal gyrus. More conservative whole-brain analyses noted activity in parts of the medial prefrontal cortex, insula, and nucleus accumbens. Although this initial study provided an important window into the neural substrates of ASMR, the authors also noted several limitations. These included the lack of a control group that did not experience ASMR as well as a small sample size ($n = 10$).

The current research builds upon the research of Lochte et al. by testing a larger number of participants and including a control group of ASMR-insensitive individuals. Individuals with ASMR and age- and sex-matched control participants viewed three popular YouTube.com videos designed to elicit ASMR tingles, as well as three control videos (i.e., videos that do not trigger ASMR tingles) of similar complexity. This design allowed us to corroborate the results of *Lochte et al. (2018)* in ASMR participants while also demonstrating that similar neural changes do not occur in individuals who do not experience ASMR. Although the current research was performed prior to the publication of Lochte et al. work, we also hypothesized that ASMR would be associated with increased activity in regions related to emotional responses including the anterior cingulate cortex, nucleus accumbens, and hypothalamus. The functions of these brain areas are consistent with the subjective reports of the ASMR experience, with anterior cingulate activity being related to both attentional control and emotion-attention interactions (*Gasquoine, 2013*), the nucleus accumbens being involved with reward responses (*Floresco, 2015*), and the hypothalamus being central to hormone release and embodied emotional responses (*Craig, 2003*). We also speculated that ASMR would be associated with increased activity in sensorimotor regions (i.e., precentral and postcentral gyri). This expectation was based both upon the descriptions of the tingling sensations as reported in previous studies of ASMR (*Barratt & Davis, 2015*; *Fredborg, Clark & Smith, 2017*), as well as the results of previous neuroimaging investigations showing that positive emotional responses to auditory stimuli—particularly music—also activated sensorimotor regions of the cortex (*Kleber et al., 2007*; see *Koelsch & Stegemann, 2012*; *Schaefer, 2017*, for reviews).

# METHODS

## Participants

The participants consisted of 17 individuals with ASMR (eight males, nine females; $M_{age} = 22.71$; $SD_{age} = 4.74$) and 17 age ($\pm3$ years) and sex-matched control participants (eight males, nine females; $M_{age} = 22.76$; $SD_{age} = 5.39$). This sample size was based on published calculations for neuroimaging studies (*Desmond & Glover, 2002*;

*Hayasaka et al., 2007*) The ASMR participants were recruited from the Winnipeg, Manitoba, Canada community via social media posts and word-of-mouth. In order to confirm that participants did in fact have ASMR, participants were asked to view two ASMR-related YouTube.com videos while in the presence of the second author. All participants confirmed that the videos triggered tingling sensations. The participants also provided information about their ASMR history and completed an ASMR checklist (*Fredborg, Clark & Smith, 2017*) which outlines various triggers that typically elicit tingling sensations in individuals with ASMR.

Controls participants were recruited from the University of Winnipeg student population. These participants also viewed the ASMR-related videos to ensure that they did not experience ASMR. Moreover, these participants were interviewed by the same investigator to determine that they had never experienced ASMR. The data from two participants were removed due to excessive movement in the scanner and were replaced by new participants; the demographic data reported above reflect the ages of the sample used in our analyses.

None of the ASMR or control participants reported any history of neurological injury or psychiatric illness. Informed, written consent was provided by all participants prior to entering the MRI scanner. Participants also underwent MR safety screening prior to scanning. This research received ethical approval from the research ethics boards of both the University of Manitoba Bannatyne Campus and the University of Winnipeg (Ethical Application Ref: HS18236; B2014:078). All participants received a remuneration of $50 CAD (approximately $40 USD).

## Procedure

After providing informed, written consent and undergoing MR safety screening, participants were brought into the MRI suite. All scans began with an initial localizer image and shimming, followed by a 7-min structural MRI scan. Participants then completed six fMRI runs. Each run lasted 5 min and 15 s and consisted of a 1-min presentation of a fixation cross, a 4-min presentation of a video, and a 15-s presentation of a fixation cross. The sound for the videos was presented via speakers in the MRI suite; all participants reported that they were able to clearly hear the six videos.

The six videos presented during this study included three videos designed to elicit ASMR tingles (hereafter "ASMR-related videos") and three videos that were not likely to elicit ASMR (hereafter "control videos"). The ASMR-related videos all involved whispering and viewing a socially emotional scene (the application of make-up, a simulated lice check, or an individual brushing someone's hair). The control videos involved normal-volume speech and were on the topics of studying suggestions, gardening tips, and make-up application instructions. The order of the three ASMR-related videos and the three control videos was randomized across participants; however, the fMRI runs were presented such that participants did not view two ASMR-related or two control videos in a row (i.e., ASMR-related and control videos were alternated).

After completing the experimental session, participants indicated whether they experienced ASMR during the viewing of any of the six fMRI runs and, if so, the intensity

(out of 10, with 10 representing the highest intensity of tingles they had ever experienced) of the resulting ASMR tingles. Importantly, participants were not told in advance which videos were expected to elicit ASMR.

## Data acquisition

Data were acquired using a three-Tesla Siemens TRIO MRI scanner (Siemens, Erlangen, Germany). Anatomical data were acquired with high-resolution T1-weighted gradient-echo images using a magnetization-prepared rapid-gradient-echo sequence with the following parameters: slice thickness = one mm; distance factor = 50%; TR = 1,900 ms; TE = 2.99 ms; in-plane resolution = 1.0 × 1.0; matrix = 256 × 256; and field of view (FOV) = 250 mm.

Functional MRI data acquisition consisted of a 5 min 15 s scan using a whole-brain echo-planar imaging sequence with the following parameters: slick thickness = three mm; distance factor = 0%; TR = 3,000 ms; TE = 30 ms; flip angle = 90°; matrix = 64 × 64; and FOV = 240 mm.

## Data analysis

The post-scan ratings of ASMR-related videos by the ASMR participants indicated that some individuals did not experience tingles in response to some of the videos. However, all ASMR participants did indicate experiencing tingles in response to at least two of the three ASMR-related videos. Therefore, for each participant, we analyzed the data from two out of the three fMRI runs involving the presentation of ASMR-related videos. If a participant reported experiencing tingles in response to all three ASMR-related videos, we selected the two fMRI runs linked with the highest tingle intensity ratings for that participant. No participants experienced ASMR-related tingles to any of the control videos. The control-video fMRI runs were yoked to the particular ASMR-related video fMRI runs used for the analyses. For example, if an ASMR participant rated ASMR videos 1 and 3 as eliciting the most intense tingles, then control videos 1 and 3 were used for that participant's within-subjects contrast.

For the control participants, we analyzed the data from the two runs used for the ASMR participant to whom they were matched. For example, if an ASMR participant reported experiencing high-intensity tingles in response to ASMR-related videos 2 and 3, then the fMRI runs for ASMR-related videos 2 and 3 were used when analyzing the data for the control participant who was age- and sex-matched with that ASMR participant.

Imaging data were preprocessed and analyzed in BrainVoyager QX software (Brain Innovation, BV, Maastricht, The Netherlands). Preprocessing of functional data included slice scan time correction, 3D motion correction, linear trend removal, and temporal high pass filtering. The functional data were coregistered to the anatomical data, and were warped to standardized Talairach space. The time-series data were dummy-coded to account for multiple runs per participant. The stimulation protocol was created to represent the timing of the video presentation and baseline. Individual data were analyzed with a single subject general linear model (GLM) analysis. These were entered into a multi-study multi-subject GLM and compiled for group analysis. A random effects ANCOVA was run. The $F$-test interaction was obtained followed by contrasts of interest;

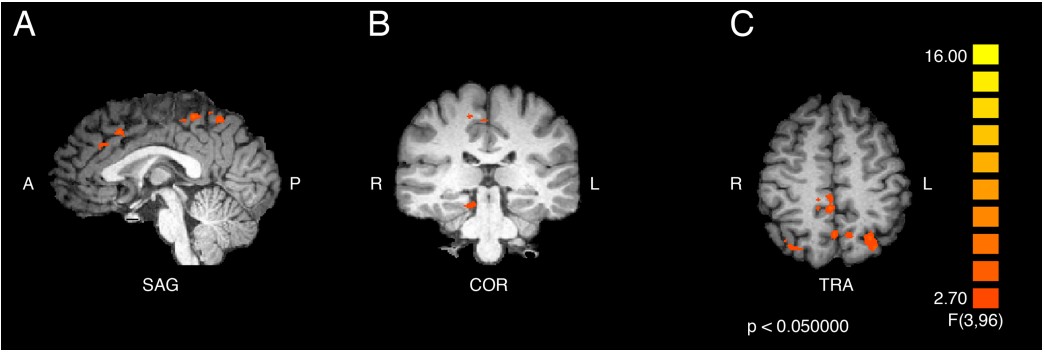

**Figure 1 Activation from the interaction of participant group and video type.** Regions of significant differential activation for the effect of the interaction between participant group (ASMR, control) and video type (ASMR-eliciting, control videos) include bilateral superior parietal, right cerebellum and paracentral lobule, and left cingulate, medial prefrontal, precentral, and inferior frontal gyri and precuneus. (A) Sagittal view. (B) Coronal view. (C) Transverse view. Results displayed at $p < 0.05$, cluster threshold corrected at 10 voxels. SAG, sagittal; COR, coronal; TRA, transverse; A, anterior; P, posterior; R, right; L, left.

specifically (11-1-10000) to address the within-subjects effect of the ASMR-related video compared to the control video in ASMR participants, (000011-1-1) to address the within-subjects effect of the ASMR-related video compared to the control video in control participants, and (1100-1-100) to address the between groups difference of ASMR participants compared to control participants while watching ASMR-related videos. Beta values were extracted and plotted for the contrasts of interest. The within-subjects results were displayed at $p < 0.01$ and corrected for multiple comparisons using a cluster threshold estimator. The interaction and between-subjects results were displayed at $p < 0.05$ and a cluster threshold of 10 voxels was applied. Clusters were converted to volumes-of-interest, and peak intensity coordinates, $t$-value, $p$-value, and voxel counts were given for each cluster. The peak intensity coordinates were entered in Talairach-Client software (http://www.talairach.org/client.html) which produced the anatomical labels and Brodmann Areas (BA), if applicable.

## RESULTS

The statistical test for the interaction effect revealed significant activity in bilateral superior parietal (BA 7), in the right paracentral lobule (BA 5) and cerebellum, and in the left cingulate gyrus (BA 32), precuneus (BA 7), inferior and medial frontal gyri (BA 9), and precentral gyrus (BA 6). The results are displayed in Fig. 1 and the peak coordinates for these regions are shown in Table 1.

Results of the within-groups contrasts are listed in Table 2 and displayed in Fig. 2 and Fig. 3. A within-groups contrast was performed comparing the neural responses of ASMR participants when they viewed ASMR-related or control videos. This contrast detected significant increases in activity in the right superior frontal gyrus (BA 8), right dorsal anterior cingulate gyrus (BA 32), and right precentral gyrus (BA 4). Significant clusters were detected in the left hemisphere in the medial frontal (BA 6), precentral (BA 6), and superior temporal (BA 22) gyri, as well as in the cuneus (BA 18). Thus, ASMR videos

**Table 1 Coordinate and statistical values for the interaction effect.**

| Region label | Brodmann area | Cluster size (# voxels) | TAL coordinates | | | f-value | p-value |
|---|---|---|---|---|---|---|---|
| | | | x | y | z | | |
| Right superior parietal lobule | 7 | 471 | 26 | −68 | 54 | 4.938689 | 0.003 |
| Right cerebellum (culmen) | * | 358 | 11 | −32 | −15 | 6.690252 | <0.001 |
| Right paracentral lobule | 5 | 697 | 2 | −38 | 51 | 9.731303 | <0.001 |
| Left cingulate gyrus | 32 | 536 | −1 | 16 | 39 | 5.823417 | 0.001 |
| Left precuneus | 7 | 434 | −1 | −59 | 48 | 5.478245 | 0.002 |
| Left medial frontal gyrus | 9 | 382 | −7 | 37 | 27 | 5.877485 | <0.001 |
| Left superior parietal lobe | 7 | 631 | −29 | −68 | 51 | 5.822857 | 0.001 |
| Left precentral gyrus | 6 | 378 | −46 | −8 | 39 | 5.686726 | 0.001 |
| Left inferior frontal gyrus | 9 | 400 | −43 | 4 | 27 | 5.402328 | 0.002 |

Note:
Regions of significant differential activation were detected for the interaction between participant group (ASMR, control) and video type (ASMR-eliciting videos, control videos). Asterisks indicate that the brain area listed on that row of the table does not have a Brodmann area.

**Table 2 Coordinate and statistical values for the contrast of ASMR-related videos > control (non-ASMR-eliciting) videos, shown for ASMR participants and control participants.**

| Region label | Brodmann area | Cluster size (# voxels) | TAL coordinates | | | t-value | p-value |
|---|---|---|---|---|---|---|---|
| | | | x | y | z | | |
| ASMR participants | | | | | | | |
| Right precentral gyrus | 4 | 343 | 50 | −8 | 42 | 3.949 | <0.001 |
| Right cingulate gyrus | 32 | 864 | 5 | 7 | 39 | 4.411 | <0.001 |
| Right superior frontal gyrus | 8 | 359 | 8 | 46 | 42 | 3.84 | <0.001 |
| Left cuneus | 18 | 445 | −19 | −86 | 18 | 3.789 | <0.001 |
| Left medial frontal gyrus | 6 | 783 | −16 | −17 | 51 | 4.44 | <0.001 |
| Left precentral gyrus | 6 | 418 | −34 | −11 | 36 | 3.637 | <0.001 |
| Left superior temporal gyrus | 22 | 398 | −58 | −47 | 15 | 3.837 | <0.001 |
| Control participants | | | | | | | |
| Right cuneus | 18 | 617 | 17 | −92 | 15 | −3.87 | <0.001 |

Note:
For the contrast involving ASMR participants, the positive t-values indicate regions of increased activity for ASMR-eliciting videos as compared to the control videos. No significant decreases in activity were observed. For the contrast involving control participants, the negative t-value indicates the region of decreased activity for the ASMR-eliciting videos as compared to the control videos. No significant increases in activity were observed for the control participants.

elicited greater activity in frontal-lobe regions as well as in areas related to vision, audition, and sensorimotor responses. The identical analysis (i.e., responses to ASMR-related or control videos) was performed using the data from control participants. This contrast yielded one significant difference: a decrease in activity in the right cuneus (BA 18).

An additional contrast was performed to compare the responses of ASMR and control participants during the viewing of ASMR-related videos. The coordinate and statistical details for the significant results are listed in Table 3 and displayed in Figs. 4 and 5. This contrast demonstrated that ASMR-related videos elicited greater activity in ASMR participants in the right paracentral lobule (BA 5), right anterior cingulate cortex (BA 24), left precuneus (BA 7), as well as bilateral activity in the thalamus (Figs. 4 and 5). Decreased

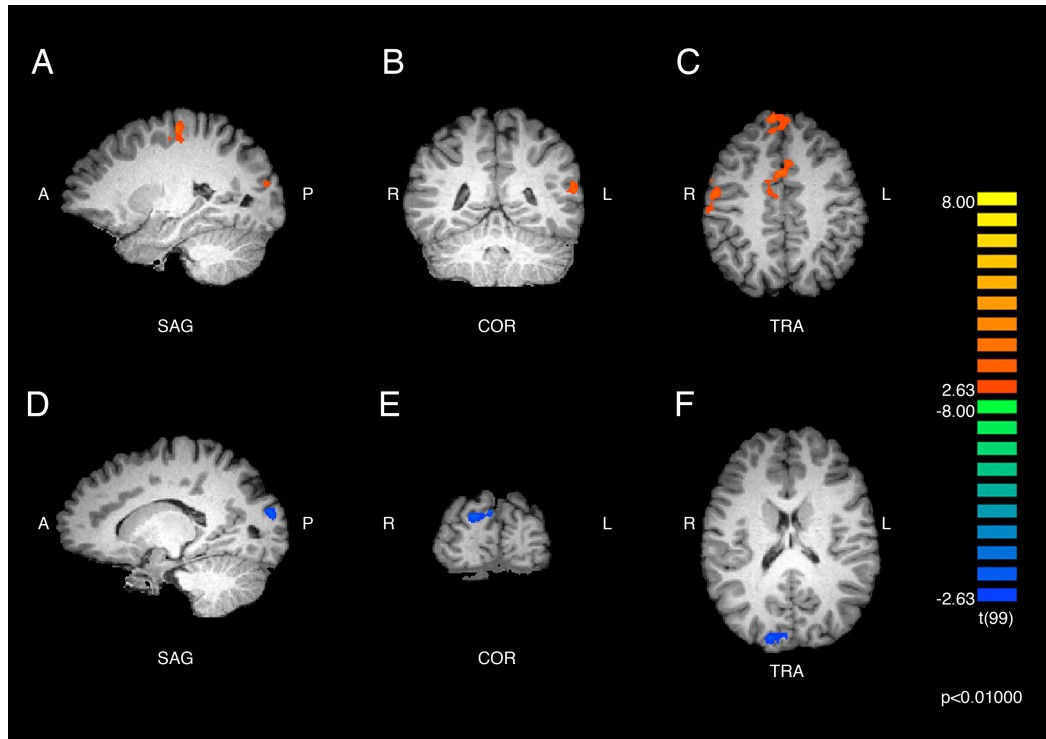

**Figure 2 Brain activity detected by the ASMR-eliciting > control video contrasts.** The results in parts (A), (B), and (C) show the responses of ASMR participants while viewing ASMR-eliciting > control videos. The peak intensity coordinates for each significantly increased cluster were located in the right superior frontal gyrus, dorsal anterior cingulate gyrus, and precentral gyrus, and the left cuneus, medial frontal, precentral, and superior temporal gyri. (A) Sagittal view. (B) Coronal view. (C) Transverse view. The results in parts (D), (E), and (F) depict the contrast showing responses of the control participants while viewing ASMR-eliciting > control videos. Control participants showed significantly decreased activity in the right cuneus while watching the ASMR-eliciting videos, as compared to watching the control videos. (D) Sagittal view. (E) Coronal view. (F) Transverse view. Contrasts are displayed at $p < 0.01$, cluster threshold corrected for multiple comparisons. SAG, sagittal; COR, coronal; TRA, transverse; A, anterior; P, posterior; R, right; L, left.

activity was observed in ASMR participants in the right lingula and the left culmen of the cerebellum.

## DISCUSSION

The current results demonstrate that ASMR has both emotional and sensorimotor characteristics. ASMR-related videos elicited activity in numerous brain areas in individuals with ASMR including medial prefrontal brain areas, the precentral gyrus, left superior temporal gyrus, and left cuneus. The same statistical contrast (ASMR videos > control videos) only elicited a decrease in activity in the right cuneus in control participants. Additionally, a comparison of responses to the ASMR videos showed greater activity in ASMR than control participants in the thalamus, left precuneus, cingulate gyrus, and medial sensorimotor regions. These data show some consistency with the previous neuroimaging study of *Lochte et al. (2018)*, namely activity in the anterior cingulate gyrus and motoric regions of the cortex; however, some of the results did differ across studies,

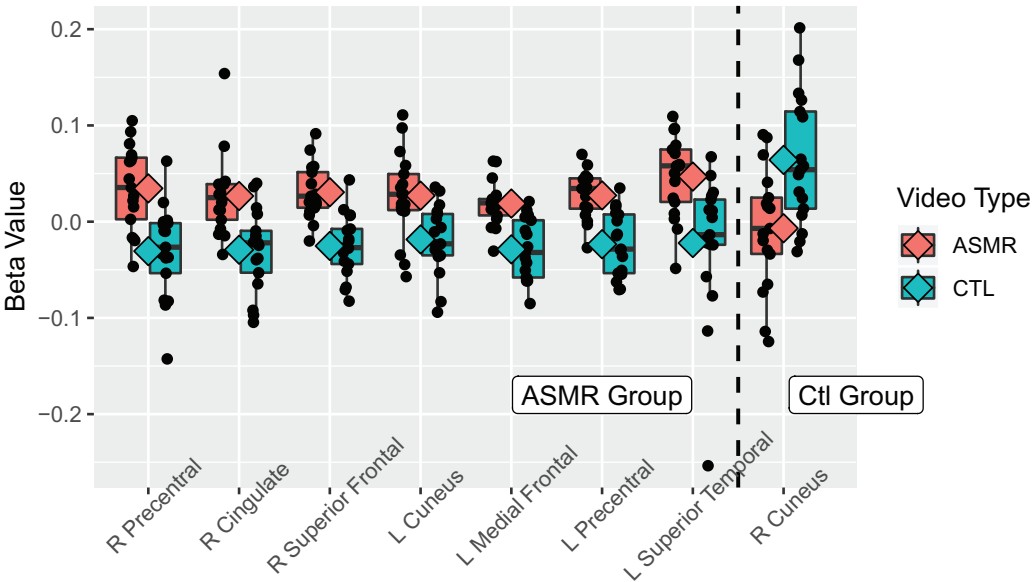

**Figure 3 Beta values for the ASMR-eliciting > control video contrasts.** Beta values for the within-subjects contrasts of ASMR-eliciting videos > control videos for the ASMR group (the left section of the plot) and for the control group (right side of plot). Video type is depicted in red (ASMR-eliciting) and green (control videos), with boxes indicating the interquartile medial range for beta values, whiskers indicating maximum, and minimum beta values (excluding outliers). The median beta value is represented by the horizontal line and the mean value by the diamond within each box. Individual beta values per cluster are represented by black circles. The anatomical label for each cluster is shown along the bottom, for reference with Table 2.

including the lack of nucleus accumbens activity in the current study. In this Discussion, we will examine the notable regions of activity detected in the current study while also highlighting avenues for future neuroimaging studies of ASMR.

The results of the current study highlight the sensorimotor nature of ASMR. Individuals with ASMR demonstrated greater activity in both the precentral (i.e., the motor cortex) and postcentral gyri (i.e., the somatosensory cortex) during the viewing of ASMR-related videos than when watching videos that did not elicit tingling sensations. Additionally, ASMR participants showed greater activity in the paracentral lobule—the continuation of the precentral and postcentral gyri on the medial aspect of the brain—during the perception of ASMR-related videos than did control participants. Together, these data demonstrate that the tingling sensations elicited by ASMR triggers are not psychosomatic; rather, they are associated with increased activity in the sensorimotor regions of the cortex.

The enhanced activity in the visual and auditory cortices may be linked with the affective nature of the ASMR videos. Previous neuroimaging research has reported increased activity in visual areas during the perception of emotional stimuli (*Morris et al., 1998*; *Surguladze et al., 2003*). Given that ASMR-related videos elicit tingling in ASMR participants, these stimuli are likely more salient and emotionally arousing than control videos. Such a difference in perceived salience and arousal may explain the activity

**Table 3 Coordinate and statistical values for the contrast of ASMR participants > control participants in response to ASMR-related videos.**

| Region label | Brodmann area | Cluster size (# voxels) | TAL coordinates | | | t-value | p-value |
|---|---|---|---|---|---|---|---|
| | | | x | y | z | | |
| Right paracentral lobule | 5 | 545 | 11 | −32 | 51 | 3.58 | <0.001 |
| Right cingulate gyrus | 24 | 378 | 14 | −5 | 36 | 3.78 | <0.001 |
| Right thalamus | * | 356 | 8 | −23 | 15 | 2.74 | 0.007 |
| Left precuneus | 7 | 323 | −7 | −56 | 51 | 3.74 | <0.001 |
| Left thalamus | * | 813 | −4 | −26 | 12 | 3.22 | 0.002 |
| Right cerebellum (lingula) | * | 641 | 8 | −47 | −15 | −3.25 | 0.002 |
| Left cerebellum (culmen) | * | 393 | −10 | −56 | −9 | −3.57 | <0.001 |
| Left cerebellum (culmen) | * | 309 | −7 | −44 | −15 | −3.13 | 0.002 |

**Note:**
Positive *t*-values indicate regions of increased activity for ASMR participants and negative *t*-values indicate regions of decreased activity for ASMR participants, as compared to control participants in response to watching ASMR-related videos. Asterisks indicate that the brain area listed on that row of the table does not have a Brodmann area.

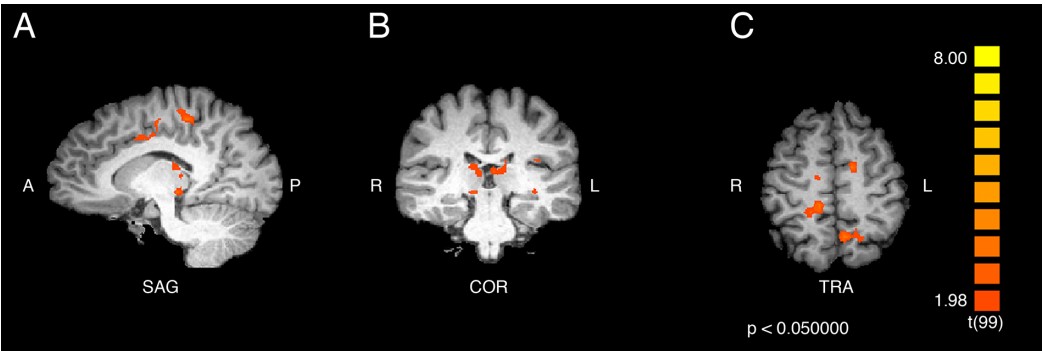

**Figure 4 Contrast showing responses of ASMR > control participants while viewing ASMR-related videos.** Activity was increased in the right paracentral lobule, right anterior cingulate cortex, left precuneus, and bilateral thalamus, and decreased in bilateral cerebellum, for ASMR participants while viewing ASMR-related videos, as compared to the control participants watching the same videos. (A) Sagittal view. (B) Coronal view. (C) Transverse view. Contrast is displayed at $p < 0.05$, cluster threshold corrected for multiple comparisons. SAG, sagittal; COR, coronal; TRA, transverse; A, anterior; P, posterior; R, right; L, left.

detected in the cuneus during the ASMR-related video vs. control video contrast and in the precuneus during the contrast of ASMR and control participants' responses to ASMR-related videos. Unfortunately, the ASMR participants' ratings of the intensity of their tingling responses did not provide much variability, thus limiting our ability to examine this hypothesis. This hypothesis could be tested in future research by asking participants to provide intensity ratings at multiple times within each fMRI run. This increase in measurements would also provide additional statistical power for analyses.

An additional potential explanation for the observed activity in the auditory cortex relates to the acoustic qualities of the ASMR videos themselves. *Barratt, Spence & Davis (2017)* found that ASMR tingles were most likely to be elicited by lower-pitched, complex sounds. It is possible that the increase in superior temporal lobe activity observed when ASMR participants listened to ASMR videos could be related to an enhanced sensitivity to

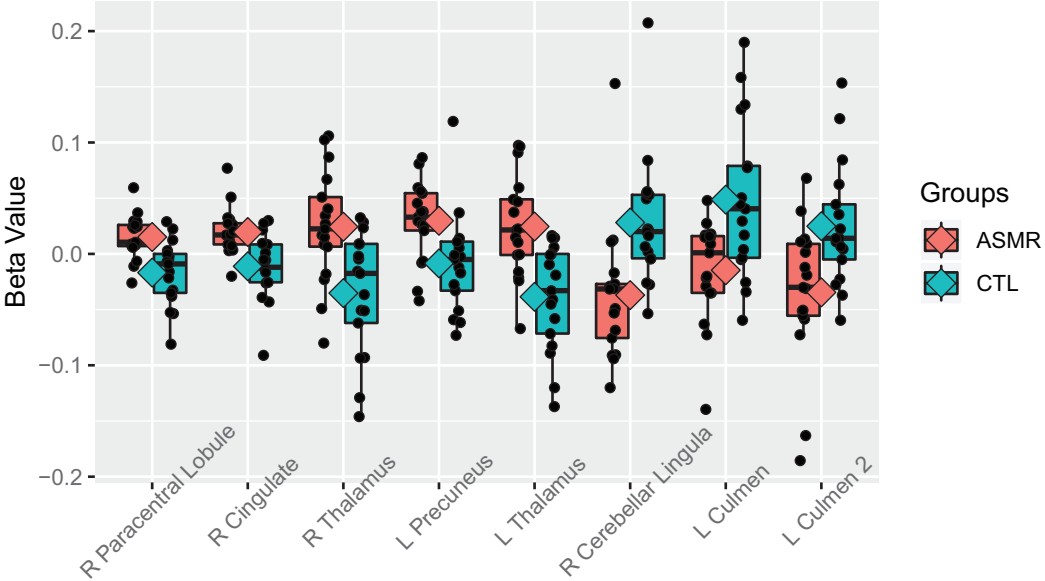

**Figure 5 Beta values of the contrast between ASMR participants and control participants while watching the ASMR-eliciting videos.** Beta values for the between-subjects contrasts of ASMR participants as compared to control participants while watching the ASMR-eliciting videos. ASMR participants are depicted in red and control participants in green. The boxes indicate the interquartile medial range for beta values, while whiskers indicate maximum and minimum beta values (excluding outliers). The median beta value is represented by the horizontal line and the mean value by the diamond within each box. Individual beta values per cluster are represented by black circles. The anatomical label for each cluster is shown along the bottom, for reference with Table 3.

these low-frequency sounds. This possibility could be tested by manipulating the acoustic characteristics of the test stimuli. Unfortunately, the current study, which only included data from two ASMR-video runs and two control-video runs for each participant, did not have enough variability in ASMR triggers to test this possibility.

The observed activity in the dorsal anterior cingulate gyrus is more difficult to classify, as this region is involved with a number of cognitive and affective functions. The significant clusters in the current study included Brodmann areas 24 and 32. These regions are part of the cognitive/attentional region of the anterior cingulate typically involved in attentional control (*Bush, Luu & Posner, 2000*). This activity seems reasonable given that ASMR-relevant stimuli would likely be quite engaging to individuals with ASMR. Moreover, it is possible that the observed activity in the dACC may be related to its membership in the salience network—a network that functions as a switch between networks involved in internal mentation (DMN) and attentional processes (central executive network) (*Goulden et al., 2014*). Given that the ASMR experience involves an attention to an external stimulus which evokes an internal sensation, it is possible that the dACC facilitates this experience in ASMR. Importantly,

this activity is consistent with the dorsal anterior cingulate activity reported by *Lochte et al. (2018)*. The Montreal Neurological Institute coordinates reported in that study (−2, 24, 38) translate into Brodmann area 8, a medial region slightly above the activity observed in the current study. Thus, there appears to be inter-study consistency related to ASMR in this brain region. It is also worth considering, however, that this region of the anterior cingulate cortex also has reciprocal connections with the more ventral, affective regions of this structure (*Hamani et al., 2011*; *Öngür & Price, 2000*). Therefore, it is premature to conclude that this anterior cingulate activity is exclusively attentional in nature.

The contrast comparing neural responses of ASMR and control participants during the viewing of ASMR-related videos also detected bilateral activity in the thalamus. Given that the thalamus serves as a "sensory relay station" for visual, auditory, and somatosensory information (*Guillery & Sherman, 2002*; *Sherman, 2017*)—the sensory modalities typically associated with ASMR—the enhanced activity in this region in ASMR participants implies that atypical sensory integration is likely occurring at this very early stage of sensory processing. This integration could take the form of thalamic nuclei that typically process stimuli from one sensory modality (e.g., somatosensation) becoming active in response to additional modalities (e.g., audition), perhaps due to overlapping synaptic projections between nuclei. It is also possible that the enhanced thalamic activity was a result of hypersensitivity in a specific thalamic nucleus (e.g., the pulvinar). Additional research is needed to test these possibilities.

The atypical thalamic activity noted in the current research is consistent with a resting-state fMRI study comparing ASMR and control participants which showed inter-group differences in the functional connectivity of the thalamus (*Smith, Fredborg & Kornelsen, 2017*). It also complements an unusual case study of a patient who acquired a form of sensory-emotional synesthesia following a thalamic stroke (*Schweizer et al., 2013*). This patient did not experience synesthesia prior to his infarct; however, during neurorehabilitation, he discovered that he experienced numerous anomalous sensory associations, including sensory-emotional synesthesia. Most notable among these was his response to brass musical instruments (particularly the brass music from older James Bond movies), which he said led to an embodied experience that was quite pleasurable. Although it is unclear whether this patient experiences ASMR per se, the phenomenological similarity between his post-stroke sensations and ASMR is noteworthy and suggests that atypical thalamic connectivity may underlie both phenomena.

A particularly surprising result from the current study was the lack of activity in reward centers such as the nucleus accumbens. This region was active in *Lochte et al. (2018)* ROI analysis of 10 individuals with ASMR. This difference may be due to the fact that several of the videos used in the previous study involved the actors touching each other (e.g., a grooming video) or to the fact that the current study measured brain activity across the entire 4-min block of the video rather than segmenting it based on participants' self-reports of tingling. In future studies, it would be useful to combine psychophysiological measurements and self-reports during the acquisition of fMRI data.

This combined methodology would allow us to more thoroughly map the central- and peripheral-nervous-system responses to ASMR triggers.

## LIMITATIONS AND FUTURE DIRECTIONS

Although the current research provides novel information about the neural responses of individuals with and without ASMR to videos designed to elicit ASMR, its shortcomings also highlight avenues for future research. Survey studies of individuals with ASMR note that whispering, tapping, and binaural sounds are common triggering stimuli (*Barratt & Davis, 2015*; *Barratt, Spence & Davis, 2017*; *Fredborg, Clark & Smith, 2017*). Thus, it would be informative to have ASMR participants and matched controls perceive these different types of stimuli while in the fMRI suite to determine whether the neural responses during ASMR experiences differ across trigger types. It would also be interesting to examine whether responses differ when socially intimate triggering stimuli (e.g., grooming videos) involve a first- vs. a third-person perspective. A third limitation relates to how the subjective intensity of the ASMR tingles were measured. It is unclear whether the self-report ratings of tingle intensity are comparable across participants (i.e., is unclear whether a 5/10 rating for one participant reflects the same ASMR intensity as a 5/10 rating from another participant). Future studies should acquire psychophysiological measures such as SCR and heart rate in order to provide a more valid measure of ASMR intensity (see *Poerio et al., 2018*). Finally, using a more temporally sensitive neuroimaging technique such as electroencephalography or magnetoencephalography would allow researchers to more precisely measure the time course of activity during the ASMR experience; specifically, these techniques could measure the location and frequency of neuronal activity immediately prior to and following the onset of ASMR tingles. Therefore, although the current study and the work of *Lochte et al. (2018)* provide complementary insights into the neural architecture of ASMR, there is a clear need for additional neuroimaging research in order to delineate the neural substrates of this atypical sensory-emotional experience.

## CONCLUSIONS

The results of the current study highlight the complexity of the ASMR experience. When viewing ASMR-eliciting videos, individuals with ASMR showed increases in neural activity in regions of the cortex related to attention, audition, emotion, and movement. This activity was not observed in control participants. When responses of ASMR and control participants viewing ASMR videos were compared, individuals with ASMR showed greater activity in the thalamus, anterior cingulate cortex, precuneus, and medial sensorimotor regions. Together, these analysis demonstrate that ASMR is not simply a sensory *or* an emotional phenomenon. Instead, the data suggest that ASMR involves sensory, motoric, affective, and attentional components.

## ACKNOWLEDGEMENTS

The authors wish to thank Teresa Figley and the radiology technical staff for their assistance with data collection.

### Funding

This work was supported by the Natural Sciences and Engineering Research Council (NSERC) of Canada (grant number RGPIN-2014-03928). The funders had no role in study design, data collection and analysis, decision to publish, or preparation of the manuscript.

### Grant Disclosure

The following grant information was disclosed by the authors:
Natural Sciences and Engineering Research Council (NSERC) of Canada: RGPIN-2014-03928.

### Competing Interests

The authors declare that they have no competing interests.

### Author Contributions

- Stephen D. Smith conceived and designed the experiments, performed the experiments, analyzed the data, authored or reviewed drafts of the paper, approved the final draft.
- Beverley K. Fredborg conceived and designed the experiments, performed the experiments, authored or reviewed drafts of the paper, approved the final draft.
- Jennifer Kornelsen conceived and designed the experiments, analyzed the data, prepared figures and/or tables, authored or reviewed drafts of the paper, approved the final draft.

### Human Ethics

The following information was supplied relating to ethical approvals (i.e., approving body and any reference numbers):

The Bannatyne Human Research Ethics Board and the University of Winnipeg Human Research Ethics Board both granted ethical approval to carry out this research (Ethical Application Ref: HS18236; B2014:078).

### Data Availability

The complete list of active voxels for the grouped analyses are available in a Supplemental File. Rows 1-3773 indicate the x,y,z location of each active voxel in the contrast of ASMR videos > Control videos. Rows 3774-3861 indicate the location of each voxel in the contrast of ASMR participants > Control participants when viewing ASMR videos.

### Supplemental Information

Supplemental information for this article can be found online at http://dx.doi.org/10.7717/peerj.7122#supplemental-information.

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
