# Peer review of "A functional magnetic resonance imaging investigation of the autonomous sensory meridian response"

_PeerJ, doi:10.7717/peerj.7122_

## Round 0.1 · original submission · Major Revisions

Your manuscript has now been seen by 3 reviewers. You will see from their comments below that while they find your work of interest, some major points are raised. We are interested in the possibility of publishing your study, but would like to consider your response to these concerns in the form of a revised manuscript before we make a final decision on publication. We therefore invite you to revise and resubmit your manuscript, taking into account the points raised. Please highlight all changes in a 'tracked changes' file.

·

Basic reporting

The method and analyses are clear throughout. My only comment here is that the hypotheses could be more thoroughly explained. Specifically, why did the authors expect to see increased activity in sensorimotor regions for ASMR participants? Several brain regions are also mentioned that are expected to be correlated as these are associated with emotional responses – are these for emotional responses in general or emotional responses we’d expect to see triggered in ASMR?

It would also strengthen the paper if the authors explained more why an imaging approach is helpful i.e., what this approach would add that we can’t get from cognitive or questionnaire-based measures.

Experimental design

The study employed an experimental design with good controls.

Validity of the findings

The findings look to be well reported and well explained.

Additional comments

I appreciate the opportunity to review this paper. This manuscript reports a study examining the neural underpinnings of ASMR using fMRI. Seventeen ASMR participants were compared to matched controls whilst watching ASMR videos and control videos. The authors find greater activation in ASMR participants to ASMR videos in areas representing emotional responses and sensorimotor activity (compared to controls). The study had a good design and built on prior work that has been done in this area but with control participants, control videos and a slightly larger sample. This is an emerging and topical area of Psychology and I very much welcome research in this area. The researchers ask a good question and the manuscript is well-written and well-reported. Good avenues for future research are given, as well as interesting speculation about the role of these brain regions in ASMR.

Minor comments for clarity:

Introduction:
-The abstract gives a good summary of the results, however, in the last line it says: ‘Together, these results highlight the fact that ASMR involves an unusual integration of activity in brain areas related to sensation, emotion, and attention’ – could the authors find a clearer word other than ‘unusual’?
The sentence: ‘These “tingles” are often accompanied by a calming feeling that many individuals report as being emotionally positive’ needs a reference (perhaps Poerio et al ?).
-On commenting on the Smith et al., (2017) study, the authors mention their findings are consistent with other studies of atypical sensory experiences. Could the authors specifically name any examples here?
-When reporting on Poerio and colleagues – could the authors add that the increase in skin conductance responses (SCR) and a decrease in heart rate was only found in ASMR participants and not controls?
-When mentioning that ASMR could also be an arousing experience, could the authors word this as ‘physiologically arousing’ to avoid any links with sexual arousal as this has been a misconception in the media?
-‘…Questionnaire-based study of the curiosity component of mindfulness’ is not clear.

Discussion:
-The authors mention ASMR participants show more activity in the ‘paracentral lobule’ but do not explain this further. What is this region responsible for? More explanation for a non-expert in cognitive neuroscience would be helpful.
-The discussion of the role of the potential role of the thalamus was really interesting. The authors mention possible ‘atypical sensory integration’ – could the authors be more specific here about they suggest may be happening here?

·

Basic reporting

The manuscript is clearly written and easy to follow. Appropriate references are provided. Raw data is not shared. Instead the statistics of the voxels in the clusters that identified using a cluster threshold estimator are provided as supplementary material.

I have some suggestions that may help to improve the structure of the manuscript:

The authors may reconsider the order of paragraphs in the introduction. The second paragraph deals with resting-state fMRI, the third paragraph with physiological changes and the fourth again with fMRI but now task-related activation. It would make sense to discuss the fMRI results together.

The rationale for the study is not entirely clear. One previous study (Lochte et al, 2018) has investigated task-related fMRI activity in ASMR, although without control participants. Was it the aim to replicate their findings using a larger sample and including a control group? If so, this should be made more explicit.

This also relates to the hypothesis that ASMR, but not control, participants would show increased activity in sensorimotor regions. Why did you expect increased activity in specifically in sensorimotor regions if Lochte et al. found increased activity in a number of other regions?

It would be helpful to start the Discussion with an overview of all results, rather than selectively starting with discussing results that align with the hypothesis. How do these results compare with the previous study by Lochte et al.? After discussing the general findings, findings in individual regions can be discussed in more detail.

Line 112: It seems the authors did not pre-register their hypothesis. I would therefore suggest to tone-down "We therefore predicted", e.g. into "We therefore expect"

Line 243: "are represented in" is rather speculative. I would suggest staying closer to the results, e.g. "are associated with increased activity in the"

Experimental design

The main analyses are a within-subject contrasts between ASMR-related and control videos performed separately in both groups and a between-subject contrast between the ASMR and control group for ASMR-related video only. The choice for these tests should be better justified. In particular, wouldn't an interaction effect between group and video be a more appropriate test?

The authors should provide some justification for the sample size used in the study.

Validity of the findings

The within- and between-subject contrast are presented in separate figures and it is not very clear to what degree these regions overlap. An additional test for interaction effects may help (see above).

It would also be insightful to present the underlying data for key areas of interest, for example, by plotting the betas in both groups and for both types of videos. Ideally, these figures would show both the group averages as well as data from individual participants (e.g. using a scatter plot).

Line 241: "The challenge for future studies is to delineate the time course of these responses" Why is this a challenge for future studies? The authors could plot the BOLD response in key regions of interest in both groups and in response to both types of videos.

·

Basic reporting

No comment.

Experimental design

No comment.

Validity of the findings

No comment.

Additional comments

This manuscript deals with the functional neural basis of the ASMR experience. The data shows increased activity in sensori-motor and affective areas of the brain in ASMR-responders.

First, please let me apologise for running late with my review. This is entirely my fault, and does not reflect on the quality of the submission.

I have no major problem with the way the study was conducted, or with the interpretation of the data. I have a few comments for clarification:

1. Line 51. 'Davies' should be 'Davis' (I get that a lot).
2. Line 192+. I am not familiar with BrainVoyager analysis. Are the contrasts able to detect decreases in activation, or are they only sensitive to increases? This is important, as I would be interested to know if ASMR-sensitive people are suppressing brain activity during the experience.
3. Is there enough data to correlate the change in activation with the subjective tingliness of the ASMR rating? That might get at the role of each brain area in mediating ASMR. The authors state (Line 254) that there is little variance in the subjects' responses, but could there be enough for a non-parametric analysis?
4. Line 247+. The authors suggest that activation in visual and auditory cortices may relate to the emotional content of the exerience. Wile this may be true, could it relate more closely to the content of the specific video? For example, more auditory activity in a more whispery video?
5. Line 259+. The authors specualte about the role of dorsal anterior cingulate cortex. Could I suggest that there may be an interesting interaction here with the salience network? The SN switches between the default mode and the central executive networks, so the dACC may be a gateway to allow ASMR-sensitive people to switch into the right state. I am an author on a paper on this topic that I confess I don't completely understand (the DCM part, at least), but it may be a good starting point if the authors are interested: https://doi.org/10.1016/j.neuroimage.2014.05.052

---

## Round 0.2 · accepted · Accept

Thank you for the revised manuscript and response letter. I am pleased to inform you that your manuscript "A functional magnetic resonance imaging investigation of the autonomous sensory meridian response" has been accepted for publication in PeerJ.

·

Basic reporting

This is clear.

Experimental design

This seems to be done to a good standard though I am not an expert in fMRI.

Validity of the findings

The analyses seem robust and statistically sound; the conclusions are based on the data and any speculation is clearly identified as such.

Additional comments

I think the authors have done a good job of revising the manuscript. Thank you for taking on board my comments.

·

Basic reporting

no comment

Experimental design

no comment

Validity of the findings

no comment

Additional comments

The authors have adequately addressed my comments and I appreciate they have added additional figures showing the interaction effect and the beta values of individual participants.

A minor point is that the labels 'ASMR Group' and 'Ctl Group' are not entirely visible in Fig. 3.